# Extensive iron–water exchange at Earth's core–mantle boundary can explain seismic anomalies

Katsutoshi Kawano[1], Masayuki Nishi [1,2] ✉, Hideharu Kuwahara [2], Sho Kakizawa [3], Toru Inoue[4] & Tadashi Kondo[1]

Seismological observations indicate the presence of chemical heterogeneities at the lowermost mantle, just above the core–mantle boundary (CMB), sparking debate over their origins. A plausible explanation for the enigmatic seismic wave velocities observed in ultra-low-velocity zones (ULVZs) is the process of iron enrichment from the core to the silicate mantle. However, traditional models based on diffusion of atoms and penetration of molten iron fail to account for the significant iron enrichment observed in ULVZs. Here, we show that the chemical reaction between silicate bridgmanite and iron under hydrous conditions leads to profound iron enrichment within silicate, a process not seen in anhydrous conditions. Our findings suggest that the interaction between the core and mantle facilitates deep iron enrichment over a few kilometres at the bottom of the mantle when water is present. We propose that the seismic signatures observed in ULVZs indicate whole mantle convection, accompanied by deep water cycles from the crust to the core through Earth's history.

At the Earth's core–mantle boundary (CMB), an environment characterised by extreme pressure and temperature, dynamic interactions between the liquid metallic core and mantle minerals occur. Seismological observations have revealed the presence of heterogeneous structures known as ultra-low-velocity zones (ULVZs) in the lowermost mantle, extending tens of kilometres in thickness. These zones exhibit a significant reduction in seismic wave velocities[1–3].

Many studies have shown that iron enrichment in mantle minerals is a significant factor contributing to the properties of ULVZs. For instance, it has been demonstrated that iron-rich mantle minerals exhibit low sound velocities[4–7]. The diminished seismic wave velocities observed in FeO-rich mantle minerals quantitatively align with those of ULVZs[8,9]. Several hypotheses have been proposed to explain the origin of iron-rich ULVZs, including a basal magma ocean[10] and the subduction of banded iron formations[11], both of which involve the increase in iron content. In addition to the hypotheses above, iron enrichment

from the core to the mantle also emerges as a compelling explanation for the unique seismic characteristics observed within ULVZs[12–14]. The gradual decrease in seismic velocities toward the CMB within ULVZs provides additional evidence supporting the notion of core–mantle interactions[15].

However, the process of iron enrichment from the core to the mantle encounters significant obstacles due to reaction kinetics. For instance, the low diffusion coefficient of iron in silicate minerals impedes iron enrichment to a degree comparable with the thickness of ULVZs, even when considering the age of the Earth[16–18]. Additionally, the concept of molten iron penetration into mantle minerals via morphological instability, as observed in the reaction between (Mg,Fe)O ferropericlase and molten iron, has been proposed[13,14]. Yet, recent experimental studies have shown no evidence of iron penetration occurring within silicate bridgmanite and post-spinel phases, which are the predominant mineral aggregates in the lower mantle[19]. Thus, both diffusion and penetration

[1]Department of Earth and Space Science, Osaka University, Toyonaka 560-0043, Japan. [2]Geodynamics Research Center, Ehime University, Matsuyama, Ehime 790-8577, Japan. [3]Japan Synchrotron Radiation Research Institute, Sayo 679-5198, Japan. [4]Department of Earth and Planetary Systems Science, Hiroshima University, Higashi-Hiroshima 739-8526, Japan. ✉e-mail: nishimasa@ess.sci.osaka-u.ac.jp

mechanisms within silicate mantle minerals fail to account for iron enrichment at a significant distance from the CMB.

Water emerges as a crucial component facilitating active chemical reactions between the core and mantle owing to its siderophile nature and exceptionally high mobility through minerals. Experimental studies suggest that surface water is transported to the deep lower mantle via hydrous phases and nominally anhydrous minerals through plate subduction[20–23]. $FeO_2H_x$ domains resulting from iron–water reactions at the CMB can effectively explain the density and seismic wave velocities of both P- and S-waves in ULVZs[24]. However, a recent study considering realistic water concentration, the unlimited availability of iron in the core and the limited water supply resulting from mantle downflow revealed that the $FeO_2H_x$ phase becomes unstable, leading to the local accumulation of FeO-rich layers at the bottom of the mantle[25]. Strong partitioning of hydrogen into liquid iron within

ULVZs, as revealed in recent studies, also suggests instability of $FeO_2H_x$ owing to hydrogen incorporation into the core[25,26].

Here, we conducted reaction experiments between bridgmanite and iron using a multi-anvil apparatus. To investigate the impact of water on iron enrichment from the core to the mantle, we compared bridgmanite polycrystals synthesised under hydrous and anhydrous conditions.

## Results and discussion
### Deep iron incorporation via iron–water exchange
The experimental conditions and results are summarised in Supplementary Table 1. Back-scattered electron (BSE) images and elemental mappings of the samples recovered after the experiments are shown in Fig. 1 and Supplementary Fig. 3. In contrast to previous experimental results indicating no reaction layer between molten iron alloy and polycrystalline silicate bridgmanite[19], our findings revealed the

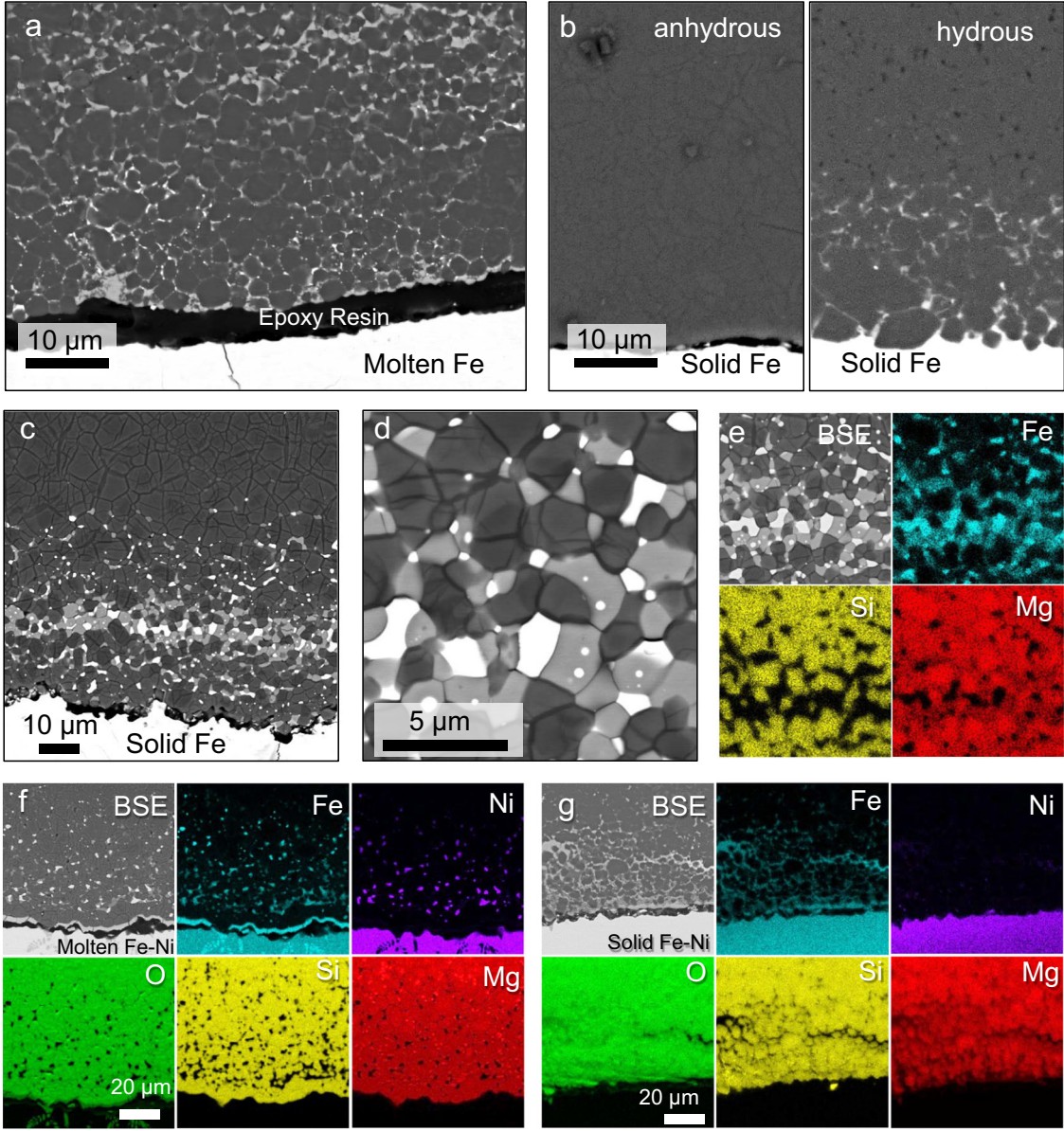

**Fig. 1 | Back-scattered electron (BSE) images and elemental mapping of run products. a** Under hydrous conditions at 2473 K for 1 min (OT2842). **b** Under anhydrous (left) and hydrous (right) conditions at 1500 K for 180 min (OT2915a,b). **c** Under hydrous conditions at 1773 K for 180 min (OT2829). **d** A magnified view of the FeO-rich layer shown in **c**. The phases present are bridgmanite (dark grey), ferropericlase (light grey), and metallic iron (white). **e** Fe, Si, and Mg distributions in **d. f**, **g** BSE images and Fe, Ni, O, Si, and Mg distributions of the recovered samples under hydrous conditions using an Fe–Ni alloy annealed at (**f**) 2473 K (OT2933b) and (**g**) 1473 K (OT2934b). Ni enrichment in the FeO-rich layer suggests penetration of the molten Fe–Ni alloy. The quenched Fe-Ni liquid forms metal dendrites due to the separation of FeO.

formation of an FeO-rich layer under hydrous conditions, comprising ferropericlase, iron-rich bridgmanite, and metallic iron (Fig. 1a). Similar reaction layers were observed at lower temperatures, even when iron was solid under hydrous conditions (Fig. 1b–e). The thickness of these layers, several tens of micrometres, exceeds the Fe–Mg diffusion length in bridgmanite at comparable pressure and temperature conditions by several orders of magnitude[16,17], highlighting the essential role of water in facilitating the reaction between iron and bridgmanite.

Fourier-transform infrared spectroscopy (FTIR) spectra of recovered polycrystalline bridgmanite before and after the iron–water exchange revealed the reduction of broad peaks in the range of 2800–3500 cm$^{-1}$, attributed to OH stretching vibrations (Supplementary Fig. 1), thus indicating the likely movement of water components into the metallic iron. In situ X-ray and neutron diffraction studies suggest that water induces iron oxidation and iron hydrogenation to form FeO and FeH$_x$ under high pressure, following the reaction[25,27,28]:

$$3Fe + H_2O = FeO + 2FeH \tag{1}$$

Compositional analysis of recovered samples (Supplementary Table 2) demonstrates that FeO generation leads to further partitioning reactions, forming Fe-bearing bridgmanite and ferropericlase, described by the equation:

$$FeO + MgSiO_3 = (Mg,Fe)O + (Mg,Fe)SiO_3 \tag{2}$$

A negligible quantity of Al for convenience was ignored in the equation. Thus, the growth of the FeO-rich layer is initiated by water-induced iron oxidation and subsequent excess FeO partitioning among iron and minerals (referred to as iron–water exchange hereafter).

Our experimental findings indicate that iron–water exchange occurred regardless of whether the iron was in liquid or solid form in the presence of water (Fig. 1). The presence of small grains of metallic iron within the FeO-rich layers may be explained by the penetration of liquid iron into the polycrystalline bridgmanite. On the other hand, metallic iron was observed even after solid–solid reactions at lower temperatures (Fig. 1e), suggesting its separation from bridgmanite through a charge disproportionation reaction of ferrous iron (3Fe$^{2+}$ to 2Fe$^{3+}$ and Fe$^0$), as Fe$^{3+}$ is stabilised in bridgmanites with a high Fe$^{3+}$/Fe$^{2+}$ ratio[29,30]. As confirmed through elemental maps (Fig. 1), neither FeSi phase nor hydrous FeOOH phases were observed.

## Mechanisms and kinetics of the reaction

According to the iron–water exchange mechanism outlined earlier, the growth of the FeO-rich layer occurs through chemical reactions between water and iron. Consequently, the ratio between the quantity of water and the contact surface area with iron (referred to as the water/interface ratio) is expected to control the thickness of the FeO-rich layer. Figure 2a illustrates the change in rim thickness of the FeO-rich layer as a function of the water/interface ratio. At temperatures where iron is solid, the rim thickness increases with the water/interface ratio. Our experimental results indicate that neither temperature nor annealing time significantly affected the thickness of the FeO-rich layer within the timescales of our experiments when the iron is solid (Supplementary Table 1).

Figure 2b presents the expected diffusion length of iron within bridgmanite and ferropericlase based on their Fe–Mg interdiffusion coefficients at 24 GPa[16,31]. The calculated diffusion length within bridgmanite at the experimental temperatures and timescales (i.e., within a few tens of hours) is ~10$^{-4}$–10$^{-2}$ μm, rendering it too slow to observe via BSE imaging. Considering such slow diffusivity, the diffusion-controlled iron enrichment within bridgmanite at the bottom of the lower mantle is calculated to be only a few meters[16]. However, our experimental findings revealed iron enrichment (FeO-rich)

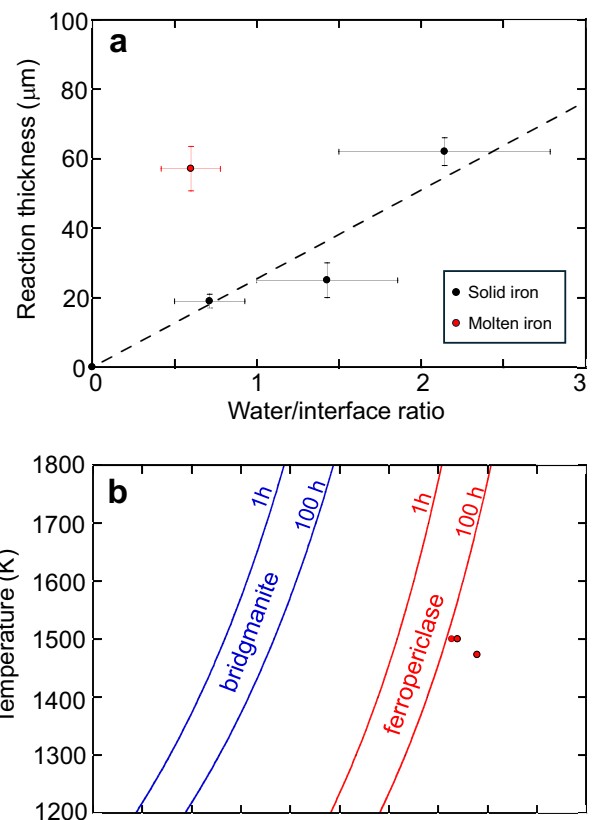

Fig. 2 | **Thickness of the FeO-rich layer. a** The thickness of the FeO-rich layer as a function of the water/interface ratio in the samples. Samples enclosed by the Au capsule without melting of Fe were selected (black symbols). The reaction thickness from samples after Fe melting is shown for comparison (red symbols). Vertical error bars indicate the standard deviations (1σ). Horizontal error bars were assumed to have 30% uncertainties, which occur due to the shape roughness of polycrystalline bridgmanite within the sample chamber. The dashed line shows the linear least square fit of the data with solid iron. **b** Fe diffusion length within bridgmanite[16] and ferropericlase[31], compared with the thickness of the FeO-rich layer (red symbols) in our recovered samples.

layers with thicknesses of ~10$^1$–10$^2$ μm (as indicated by the three symbols in Fig. 2b), several orders of magnitude larger than the Fe diffusion length within bridgmanite at comparable pressure and temperature conditions[16]. Such substantial iron enrichment could be feasible only if Fe–Mg interdiffusion in ferropericlase controlled the reaction (red lines in Fig. 2b). Additionally, grain boundary diffusion may support the progress of the reaction, as suggested by microtextures observed in Fig. 1.

Despite the limited solubility of oxygen in solid metallic iron, liquid iron can absorb a significant quantity of FeO under pressure[32,33]. Consequently, the thickness of the FeO-rich layer is expected to decrease when iron melts. However, contrary to this simple estimation, the thickness of the layer significantly increased upon iron melting (Fig. 2a). Furthermore, when we utilised the Fe–Ni alloy as the starting material and melted it, its particles migrated into the FeO-rich layer while retaining their original composition (Fig. 1f). In contrast, the flux of Ni from the solid metal to the FeO-rich layer via diffusion was limited due to the higher siderophility of Ni relative to Fe (Fig. 1g)[34]. These observations suggest the occurrence of liquid metal penetration into the FeO-rich layer, a phenomenon not observed within bridgmanite-rich domains[19]. Although direct comparison with the previous study[13] is difficult due to different experimental conditions and the limited ferropericlase fraction in our study, we observed iron enrichment with

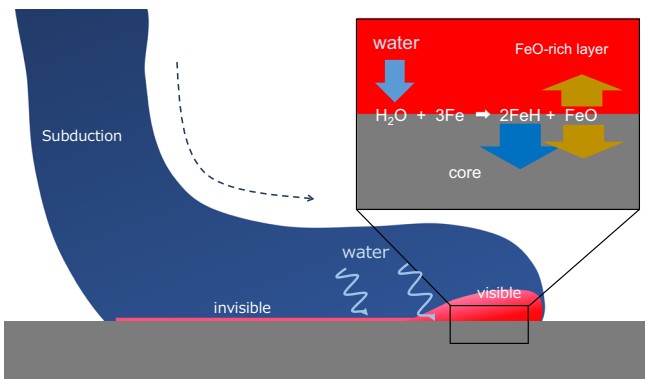

**Fig. 3 | Schematic diagram of ULVZ formation through iron–water exchange.** Deep mantle convection may deliver some quantity of water to the CMB. During prolonged heating at the CMB, the water diffuses to the outer core because of its siderophile nature and active diffusivity, resulting in an increase in the FeO component at the CMB. FeO-enriched ULVZs are formed by the partitioning of excess FeO into the core and mantle after heating and are detectable locally.

a thickness of approximately 100 μm, which is comparable to the previous findings. Thus, the increase in the ferropericlase fraction through the iron–water exchange facilitated the penetration of liquid iron, thereby significantly influencing the reaction mechanism and its kinetics.

### Effects of iron–water exchange on seismic properties

The iron–water exchange mechanism involves a combination of two distinct processes (Fig. 3). The first process entails the chemical reaction between $H_2O$ and Fe, resulting in the formation of FeO at their interface. Previous experiments have demonstrated the occurrence of this reaction under a wide range of pressures, including conditions representative of the CMB[25,27]. The second process involves the common partitioning of FeO between mantle minerals and the metallic core after the initial reaction[32]. Thus, both processes are applicable to the interaction between the mantle and core.

Recent investigations suggest that water exhibited a higher affinity for partitioning into molten iron than silicate melt during the early stages of Earth's evolution[26,35]. This characteristic of water is likely accentuated at the present CMB owing to its pressure dependence[26] and mantle solidification. Consequently, deep water cycles facilitated by mantle convection through hydrous or nominally anhydrous phases cause continuous iron–water exchange throughout Earth's history. Seismological observations imply that some ULVZs are related to plate subducted to the deep lower mantle[36,37]. This supports our hypothesis that iron-rich layers can form through the transport of water by subducted slabs and their reaction with the outer core.

Figure 4 illustrates the density, P-wave velocity ($V_p$), and S-wave velocity ($V_s$) of FeO-enriched ULVZs at CMB conditions (136 GPa and 4000 K) as a function of water content (See "Methods"). This estimation considered a pyrolytic lowermost mantle with a volume fraction of 80% $(Mg,Fe)SiO_3$ post-perovskite (or bridgmanite) and 20% $(Mg,Fe)O$ ferropericlase as dominant minerals. Mass balance calculations with a fixed ULVZ volume ($3.65 \times 10^8$ km³) predicted changes in mineral proportions and compositions within ULVZs. The supply of $H_2O$ to the core increased the FeO component and augmented the volume fraction of ferropericlase, leading to the formation of FeO-enriched ULVZs (Fig. 4a, b). Consequently, density increased while $V_p$ and $V_s$ decreased as a function of input water content (Fig. 4c–f), which explains the seismic structure of the ULVZs.

The significant reduction in $V_s$ relative to $V_p$ in some ULVZs suggests the occurrence of partial melting, a phenomenon commonly known in ULVZs[38,39]. Iron enrichment via iron–water exchange may trigger partial melting in ULVZs, as increased FeO content substantially

lowers the melting temperature of ferropericlase[14]. Our results indicate that a total water mass of approximately $3 \times 10^{20}$ kg, combined with 5 vol.% partial melting of ferropericlase, adequately explains the seismic structure of ULVZs, including the reduction in $V_p$ and $V_s$ and the increase in density. This water mass corresponds to ~1/5 of the mass of water in the oceans. Further experimental studies on sound velocity measurements of ferropericlase with different Fe components, along with seismological studies related to ULVZ volume fractions including the discovery of new ULVZs, will contribute to better water quantification.

As discussed, the high Fe diffusivity in ferropericlase relative to silicate minerals drives the growth of FeO-rich regions. However, even considering the rapid diffusivity in ferropericlase-rich domains, achieving deep iron enrichment over a few kilometres requires extremely high temperatures (>3000–4000 K) at CMB pressure[31,40,41]. Consequently, the detectability of FeO-rich layers, typically >5 km in thickness, may be localised, only becoming apparent after a sharp increase in subducted unit temperature (Fig. 3). Hence, both water content and temperature gradients around the CMB are crucial factors in ULVZ formation, contributing to their complex distribution. ULVZs may contain some metallic iron due to molten iron penetration and charge disproportionation reactions, as observed experimentally. This iron may form $FeH_x$ and FeO through additional water from the mantle, further aiding the growth of FeO-rich layers.

Through iron–water exchange at the CMB, hydrogen and oxygen are simultaneously incorporated into the outer core alongside FeO enrichment in the mantle. Mass balance calculation suggests that the iron–water exchange that forms the volume of ULVZs does not significantly affect the composition of the whole outer core. However, if we assume a thin, stable layer separated by composition at the topmost outer core, an increase in oxygen and hydrogen by a few per cent in the layer can be expected. As highlighted by a recent study[42], the incorporation of hydrogen into the core offers a plausible explanation for the formation of a low-velocity layer of outermost core[43,44]. This alteration in the composition of the uppermost outer core becomes reasonable when considering the dynamic reaction mechanism involved in the iron–water exchange. Thus, the formation of ULVZs via iron–water exchange can be linked to the enigmatic seismic structures observed in the outermost core. We conclude that whole mantle convection coupled with deep water cycling[20–24] offers reasonable explanations for the seismic structure observed at the core–mantle boundary.

## Methods
### Synthesis of bridgmanite under anhydrous and hydrous conditions

Two types of polycrystalline bridgmanite were synthesised as starting materials under hydrous and anhydrous conditions in gold capsules utilising a 3000-tonne multi-anvil apparatus (ORANGE-3000) at Ehime University. The sample assembly was similar to that reported previously[45]. The hydrous sample was synthesised at 27 GPa and 1900 K directly from a mixture of MgO, $SiO_2$, $Al_2O_3$, and $Mg(OH)_2$ brucite, aiming for a target composition of $MgSi_{0.95}Al_{0.05}H_{0.05}O_3$. Conversely, the anhydrous sample was synthesised from $MgSiO_3$ glass at 25 GPa and 1700 K. The glass was derived from a mixture of MgO and $SiO_2$, melted in a high-temperature furnace. XRD spectra confirmed a single phase of bridgmanite in both samples synthesised under anhydrous and hydrous conditions.

FTIR spectra of the polycrystalline bridgmanite synthesised under hydrous conditions exhibited broad peaks in the range of 2800–3500 cm⁻¹, attributed to OH stretching vibrations (Supplementary Fig. 1). Using the method described by Paterson[46], the water content of the sample was estimated to be 0.51(11) wt.%, consistent with the quantity of $Mg(OH)_2$ brucite mixed into the starting oxides. Apart from potential hydrogen substitution mechanisms in

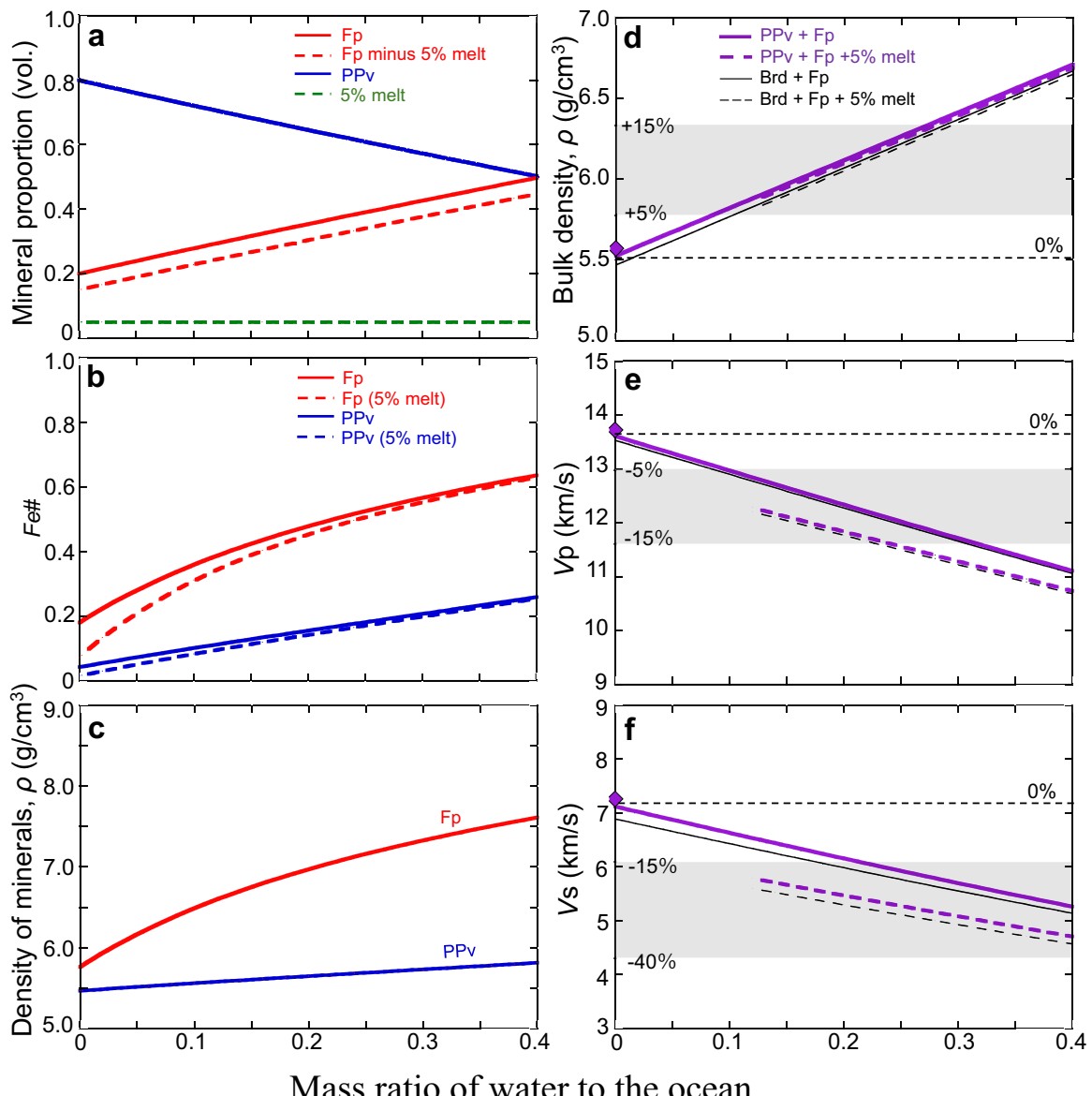

**Fig. 4 | Influence of iron–water exchange on seismic velocities as a function of water. a** Changes in the proportion of minerals in ULVZs through iron–water exchange. Dashed lines indicate the values corresponding to the occurrence of 5 vol.% partial melt. **b, c** Changes in the composition (Fe#, Fe/(Mg+Fe)) and density of minerals. **d–f** Change in density ($\rho$), $V_p$, and $V_s$. Diamonds show the values from bridgmanite[47–49], such as $Al^{3+} + H^+ = Si^{4+}$ and $2H^+ = Mg^{2+}$, trace quantities of superhydrous phase B (indiscernible from XRD) may also contain water in the polycrystalline structure, as indicated by similar FTIR peaks observed with this phase[50]. Utilising this hydrous starting sample, we conducted reaction experiments between bridgmanite and iron under hydrous conditions.

the Preliminary Reference Earth Model (PREM). Shaded areas indicate the changes in $\rho$ (+5– + 15%), $V_p$ (−5– − 15%), and $V_s$ (−10– − 40%), which are typically observed in ULVZs[3]. The mass of the ocean corresponds to $1.38 \times 10^{21}$ kg. PPv, post-perovskite; Brd, Bridgmanite; Fp, ferropericlase. Source data are provided as a Source Data file.

### Reaction experiments

The cell assembly employed for the reaction experiments is depicted in Supplementary Fig. 2. An Re sleeve served as the heater, surrounded by a LaCrO₃ thermal insulator, with gold or MgO used as sample capsules. Temperature monitoring was facilitated using W3%Re–W25%Re thermocouples. WC cubes with a truncated edge length of 4 mm served as second-stage anvils. Pure Fe or Fe–Ni alloy foils, with a thickness of 100 µm, were brought into contact with the synthesised bridgmanite within the capsules. Water content was approximately controlled by the space volume of the sample chambers. Accounting

for the surface roughness of the polycrystalline bridgmanite, the volume of bridgmanite was estimated to be half the volume of the sample chamber in calculations. The samples were compressed to 25 GPa at room temperature and then heated to temperatures ranging from 1473 to 2473 K. In runs OT2915 and OT2933, temperatures were estimated using a power-temperature relation due to instability in monitored temperatures above 1200–1400 K.

Quenched recovered samples were polished perpendicular to the sample interface. The microtexture of the reaction rim, elemental maps for Fe, Mg, Si, Ni, and O, and the chemical compositions of the present phases were acquired using a field-emission scanning electron microscope (FE-SEM, JSM-IT500HR) with an energy-dispersive spectrometer (EDS).

### Estimation of the seismic structure of the FeO-rich layer

The iron–water exchange initiates through a chemical reaction between $H_2O$ and Fe, yielding FeO and $FeH_x$. The reaction can be

expressed as follows:

$$H_2O + 3Fe = X FeO_{[mantle]} + (1-X) FeO_{[core]} + 2 FeH_{[core]} \quad (3)$$

where $X FeO_{[mantle]}$ represents the FeO component partitioned into the mantle, and $(1-X) FeO_{[core]}$ and $FeH_{x[core]}$ denote the FeO and FeH components dissolved into the core, respectively. For convenience, we fixed the $X$ value at 0.5 in our calculations, as per previous studies[32,51,52]. From Eq. (3), the FeO component within the ULVZs increased and the corresponding amount of ferropericlase and post-perovskite in the ULVZs were subtracted according to their respective volume ratios.

We assumed a simplified pyrolytic lowermost mantle comprising 80% volume fraction of $(Mg,Fe)SiO_3$ post-perovskite (or bridgmanite) and 20% $(Mg,Fe)O$ ferropericlase as dominant minerals[53], which approximately reproduces the density and sound velocities of the Preliminary Reference Earth Model (PREM). To assess the effects of the increased FeO on the mineral proportions and compositions of minerals in ULVZs (Fig. 4a, b), we employed mass balance calculations:

$$V_{ulvz} = \frac{W_{fp}}{\rho_{fp}} + \frac{W_{ppv}}{\rho_{ppv}} \quad (4)$$

$$\rho_{fp} = \left(1 - Fe\#_{fp}\right)\rho_{MgO} + Fe\#_{fp}\,\rho_{FeO} \quad (5)$$

$$\rho_{ppv} = \left(1 - Fe\#_{ppv}\right)\rho_{MgSiO3} + Fe\#_{ppv}\,\rho_{FeSiO3} \quad (6)$$

where $V_{ULVZ}$ represents the fixed volume of ULVZs ($3.65 \times 10^8$ km$^3$), based on a thickness of approximately 20 km over about 12% of the surface at the CMB[3,11]. The partition coefficient $D$ of FeO between post-perovskite and ferropericlase ($(Fe/Mg)_{ppv}$ / $(Fe/Mg)_{fp}$) is fixed at 0.2 in the calculation. In addition, we performed calculations using $D$ values of 0.05 and 0.5 to explore the sensitivity of our results to variations in FeO partitioning[54–56] (Supplementary Fig. 4). $W_{fp}$ and $W_{ppv}$ are the weight of ferropericlase and post-perovskite in the system, respectively. $Fe\#_{fp}$ and $Fe\#_{ppv}$ denote the Fe/(Fe+Mg) values in ferropericlase and post-perovskite, respectively, with an initial $Fe\#_{fp}$ value of 0.18 used in our calculation. $\rho_{MgO}$, $\rho_{FeO}$, $\rho_{MgSiO3}$ and $\rho_{FeSiO3}$ represent the densities of minerals with hypothetical compositions at 135 GPa and 4000 K.

The iron–water exchange increases the FeO contents of $(Mg,Fe)O$ ferropericlase and $(Mg,Fe)SiO_3$ post-perovskite or bridgmanite (Fig. 4b). It is known that an increase in Fe/(Mg+Fe) in ferropericlase from 0.2 to 0.4 decreases its melting temperature by at least a few hundred kelvin[14], potentially crossing the CMB temperature threshold and triggering partial melting in ULVZs.

Figure 4c–f and Supplementary Figs. 4, 5 summarise $\rho$, $V_p$, and $V_s$ of FeO-enriched ULVZs as a function of input water, based on changes in the mineral fraction and composition at 135 GPa and 4000 K. The references used for the sound velocities of each phase are detailed in Supplementary Table 3. We have used the reference values taken from the theoretical calculations in the previous study[6,7]. Linear compositional dependence on the elastic parameters of given phases was assumed. This estimate roughly reproduces the $V_p$ and $V_s$ obtained from experiments with a composition of $Fe_{0.78}Mg_{0.22}O$ (Supplementary Fig. 6), which is the closest available experimental data to the required composition range[8,9,57]. Velocities of bulk rock are made by Voigt-Reuss-Hill average. The effect of 5% partial melting on FeO-enriched ULVZs (dashed lines in Fig. 4a, b) was calculated based on the equilibrium geometry model[58], assuming a dihedral angle of 20°. The melt composition $((Fe_{0.72}, Mg_{0.28})O)$ and density (7.5 g/cm$^3$) were fixed[59].

## Data availability

The data supporting the main findings of this study are available in Supplementary Information. Source data are provided as a Source Data file. Source data are provided with this paper.

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

## Acknowledgements

We thank Y. Nishihara, T. Shinmei, and T. Irifune for their assistance with the experiments. We are also grateful to T. Tsuchiya and H. Dekura for their useful comments. This work was supported by MEXT/JSPS KAKENHI Grant numbers JP 22H01322 to M. Nishi. This work was also supported by the Joint Usage/Research Center of PRIUS, Ehime University, Japan.

## Author contributions

M.N. conceived the idea and designed the study with K.K, and T.K. K.K. and M.N. carried out high-pressure experiments. K.K., M.N., and H.K. conducted chemical composition analysis. T.I. and S.K. synthesised the polycrystalline bridgmanite under the hydrous condition. K.K. and M.N. wrote the manuscript. All authors contributed to the discussion of the results and the revision of the manuscript.

## Competing interests

The authors declare no competing interests.
