## [Peer Review File · Nature Communications]

REVIEWER COMMENTS

Reviewer #1 (Remarks to the Author):

Review of the manuscript by Kawano et al.

Kawano et al. performed the high P-T experiments at 25-27 GPa and 1473-2473K under hydrous and anhydrous conditions to try explaining ULVZs at CMB. I like the idea, but their implication strongly relies on overmuch estimation, assumption and incorrect citation, e.g., extrapolating the experimental data at 25 GPa to CMB (~135 GPa), partitioning coefficient of Fe between post-perovskite and ferropericlae, V_p and V_s of Brg (or PPv). I cannot recommend this manuscript for publication in Nature Communications. My comments/suggestions see the below:

One of main assumptions is that the reactions of $3\text{Fe} + \text{H}_2\text{O} = \text{FeO} + 2\text{FeH}$ and $\text{FeO} + \text{MgSiO}_3 = (\text{Mg,Fe})\text{O} + (\text{Mg,Fe})\text{SiO}_3$ must happen when water meets Earth's core. But, the exsolved ferrous (Fe^{2+}) could undergo disproportionation ($\text{Fe}^{2+} \rightleftharpoons \text{FeO} + \text{Fe}^{3+}$) at CMB conditions. If that, how does Fe^{3+} affect your assumed reaction? Can it still efficiently work on 'Fe diffusion' to Brg or Fp?

Nishi et al. (2017) experimentally showed pyrite-type FeOOH stabilizing at the CMB conditions, which is confusing to me why the authors claimed $\text{Fe} + \text{H}_2\text{O} \rightarrow \text{FeO} + \text{FeH}$, rather than FeO_2Hx in this study. Actually, the reaction of $\text{FeO} + \text{H}_2\text{O} \rightarrow \text{FeO}_2\text{Hx} + \text{FeHx}$ should be the result, which has been studied and used to explain ULVZs (e.g., Liu et al., 2017 Nature). Therefore, 'the idea' is not new anymore. Again, the experimental conditions in this study (25-27 GPa) are far away from the stable field of FeO_2Hx (>~90 GPa), where the authors tried to make the implication. How is possible to tell a story happening at CMB whereas using the data from >2000 km shallower?

Their estimations of seismic velocities of Brg (or PPv) could be totally wrong in Fig. 4c,d and Table S3. Their velocity result fully depends on the extrapolation of sound velocities data of olivine at the extremely low P-T conditions compared to the CMB's (Zhang et al., 2016). I cannot imagine how big error at their estimations could be because of the different mineral phases, different P-T conditions. It would be much more convincing if the authors could provide the experimentally measured velocity data of Brg+Fp at their targeted P-T conditions.

Lines 299-300: "D is the partition coefficient of FeO between post-perovskite and ferropericlae ($(\text{Fe/Mg})_{\text{ppv}} / (\text{Fe/Mg})_{\text{fp}}$), fixed at 0.4 in the calculation." In general, D ranges at ~0.4-0.9 at lower mantle P-T conditions (Irifune et al., 2010; Prescher et al., 2014; Sinmyo and Hirose, 2013), but a mantle heavily enriched in Fe^{2+} would have a D of <0.1 (Nakajma et al., 2012). The implication of this ms actually fits better with the second case – D should be much lower than 0.4. What will the results be if the authors apply D of 0.1 or 0.9? Still matching the volume of ULVZs?

Lines 109-111: What is the effect of pressure on the rim thickness of the FeO-rich layer? Will it become thinner or thicker at the CMB conditions than your estimation from the experimental conditions at the topmost lower mantle?

Lines 112-114: If I understand Table S1 correctly, T does significantly affect the thickness of the FeO-rich layer, which is different with the description from the authors.

Reviewer #2 (Remarks to the Author):

Kawano and co-authors present a series of experiments on bridgmanites interacting with iron(-nickel) metal. By using identical experiments that directly compare the effect of water in bridgmanite, as well as the effect of solid or liquid iron metal, the authors are able to make inferences on how water may affect reactions at Earth's core-mantle boundary. In particular, they show evidence that solid anhydrous bridgmanite will not react with iron, while the presence of water seems to induce the formation of FeO and FeH through reaction of water with iron. The production of FeO in the form of iron-rich (Mg,Fe)O seems to further promote diffusion of iron metal that otherwise would not occur. By varying the amount of water (or rather, volume of the hydrous bridgmanite sample), the authors demonstrate that the extent of FeO formation and iron penetration into the silicate is controlled by the amount of water available for reaction. Thus, the presence of water may promote iron enrichment of the mantle base, providing a plausible origin scenario for observed seismic structures like ultralow velocity zones that could be explained by iron-rich rock.

I believe this article would be well suited for publication in Nature Communications after several aspects are addressed and revised. Previous experiments have considered some similar reactions, especially looking at anhydrous silicates, anhydrous oxides, and less realistic water-saturated scenarios. This work is novel in its direct comparison of hydrous with anhydrous silicate material, which demonstrates a clear difference in rock-metal interaction and does so for a relatively realistic water concentration. Repetition of experiments and reproducibility of the findings is assuring to see. The scientific topic is an active hot topic in Earth science, with many open questions and ongoing work on the nature of the mantle-core interface and the role of water, especially in relation to heterogeneous mantle base structures that are actively being discovered by seismic studies. This work is a timely and valuable contribution to these discussions that I expect will motivate further experiments, especially at higher pressures. I identify a few key areas that need revision, particularly on showing more data, including some missing literature references, and reworking the interpretation for sound velocities, as well as some questions that need to be clarified by the authors before the work can be acceptable for publication.

1. Chemical analysis

(a) I am very interested to see more elemental maps reported. I do not see any maps for Mg or O. This makes it difficult to assess the claim that one single phase of iron-rich (Mg,Fe)O is being produced along with Fe metal. This is very important because this claim is the central point of the paper. How can the authors be sure that the bright grains in Fig. 1 are iron metal alloy and not some other iron-rich phase? Can the authors provide the full set of elemental maps that were collected for the heating runs that are shown in Fig. 1? What about for other experiments, such as the one shown in Table S2? It would be helpful to see all the elemental maps for that experiment (2829). See more on this below (comment 1d).

(b) In addition, I am curious to hear discussion on hydrous Fe-O-H phases that have been previously reported (e.g., ref 14 – Liu et al. 2017 Nature) and to have the authors comment on whether they think the production of such phases is possible in these samples. In other words, without the use of XRD, how can the authors be sure that the new phases formed are iron metal alloy and iron-rich (Mg,Fe)O and not some other phases? How can the authors be sure that phases like exotic Fe-O-H phases and FeSi (e.g., Knittle and Jeanloz 1991 Science, Fu et al. 2023 Nature) are not being formed?

(c) My understanding is that EPMA should have some finite activation volume in the sample usually at least ~1 μm ? This would mean that the probe is sensitive to a volume of sample that likely includes grains deeper inside the sample below the surface. Are the uncertainties on compositions reported in Table S2 realistic?

(d) Is there a reason that compositional data is provided for only one sample? It would be very helpful to see compositions for multiple samples, especially those shown in Fig. 1. This would clarify how consistent the resulting compositions of (Mg,Fe)O and bridgmanite are. In particular, does the iron content of the (Mg,Fe)O phase that is produced depend on the extent of the reaction / amount of water? Further, can the authors report an Fe partitioning coefficient between bridgmanite and magnesiowüstite across the range of samples for which compositions are measured? How do the calculated coefficients compare to previous work by Dorfman et al. (2021) Minerals?

2. Missing references

(a) The authors motivate the paper by stating that iron enrichment of mantle minerals is a strong candidate for ULVZs (lines 44-47). However, there are no references included for this in the introduction or anywhere else. There is a whole body of recent work about iron enrichment in ULVZs that is completely absent from the manuscript. This should be corrected in the introduction, which I will comment on here, and then plays into the analysis of sound velocities at the end of the paper, which I will comment on below.

Papers that should be included at minimum: Wicks et al (2010) GRL; Wicks et al. (2017) GRL; Lai et al (2022) G3. These reported low sound velocities for iron-rich (Mg,Fe)O (Wicks) and demonstrated that such FeO-rich solid rocks can explain ULVZ velocities (Lai). Other papers to be considered are Dobrosavljevic et al. (2019) Minerals (effect of iron content and methods of sound velocity modeling) and Jackson and Thomas (2021) AGU Monograph (broad range of ULVZ locations).

(b) Other mechanisms have been proposed for production of iron enrichment at the mantle base. These include in particular a basal magma ocean (Labrosse et al. 2007 Nature) and subduction of banded iron formations (Dobson and Brodholt 2005 Nature). These should at least be mentioned in the introduction as another possible way to further enrich the mantle base in iron, beyond the mechanism proposed in this study. Authors should also include Knittle and Jeanloz (1991) Science as an earliest experiment proposing iron enrichment at the CMB due to reaction of silicates with the core.

3. Sound velocity calculations

(a) The choices made for sound velocity calculations are not very clear to me. I understand the point about mass balance due to differing amounts of water. However, the elastic properties used for (Mg,Fe)O

are listed only as those calculated computationally for pure MgO and for (Mg_{0.75}Fe_{0.25})O. This is a much lower iron content than what the authors themselves found experimentally, where the experimental reaction produced an (Mg,Fe)O composition with 65% Fe. This experimental value is much closer to the iron-rich (Mg,Fe)O compositions proposed previously to explain ULVZ signatures (see reference list in my comment 2), such as iron levels of 75%-90%. In addition, those compositions have experimentally determined sound velocities and densities. So it is not clear to me why those values are not being used for example calculations. It's further not clear what the compositions of (Mg,Fe)O are in the calculations for high amounts of water content. Do these exceed the Fe=25% level? It seems they must given how low the calculated bulk sound velocities are. In that case, it is likely that a linear extrapolation from MgO and (Mg_{0.75}Fe_{0.25})O is no longer valid at higher Fe contents. Our community does not have a good understanding, if any, of iron content effect on sound velocities at intermediate compositions between Fe~25% and Fe~75%. If the authors want to model iron-rich (Mg,Fe)O sound velocities above the Fe=25% compositions, then I believe they should in some way use the experimentally reported sound velocities for very iron-rich (Mg,Fe)O as in the references above. See a suggestion below in comment 3c.

(b) Further, it is not clear from the methods how exactly the sound velocities are calculated. The authors state in line 308 that "Linear compositional dependence on the elastic parameters of given phases was assumed." There is not enough detail provided. Is this being done for elastic moduli? What assumptions are being made for stress and strain partitioning? Calculating velocities of bulk rock requires decisions on how to average elastic moduli, as in the common Voigt and Reuss bounds or the Voigt-Reuss-Hill average. Authors may refer to calculations detailed in e.g., Dobrosavljevic et al., (2019) Minerals and Lai et al. (2022) G3. This point is especially important for rocks comprised of minerals with very different elastic moduli, as in bridgmanite and iron-rich (Mg,Fe)O. In such a case, the difference between Voigt and Reuss bounds (or uncertainties on V-R-H averages) can be incredibly large (again, refer to e.g., Lai et al. (2022) G3). These details need to be explicitly stated at least in the supplementary material if not in the Methods.

(c) It seems to me that the goal of the authors is to show that reasonable amounts of subducted water can produce a high level of iron enrichment through reaction with the core and explain ULVZ properties. In light of my comments above, I think a stronger approach (or at least a strong complement to the current approach) would be to graphically show how much iron enrichment in (Mg,Fe)O is produced due to such a reaction. If the authors can plot (Mg,Fe)O composition and concentration of (Mg,Fe)O in the bulk rock as a function of reacted water, then the authors could simply refer to existing literature on FeO-rich rocks as an explanation for ULVZs (see reference list in my comment 2 above). This would be a much cleaner message in the discussion and in Figure 4 without needing to go into the many details of sound velocity modeling done in previous work. Authors can also consider plotting the x-axis as units of fraction of ocean mass instead of in units of kg (as mentioned in line 168), which are hard to interpret.

4. Discussion around CMB temperature and melting in the context of ULVZs: recent seismic studies are increasingly finding evidence for ULVZ existence in areas of active subduction where temperatures are presumed to be relatively low (e.g., Su et al. 2024 Nature Geoscience, Wolf et al. 2024 Nature Geoscience). I think these studies actually strengthen the findings of this paper. Instead of focusing the discussion so much on high temperature and partial melting, the authors could consider discussing how

their findings actually provide a compelling mechanism to create iron enrichment in regions of subducted slab debris by reaction of hydrous silicate with core fluid. My question to the authors here: do the experimental findings actually require higher temperatures to promote enough reaction for iron enrichment? Or can the authors apply the results even to colder regions similarly well to give a viable mechanism for CMB iron enrichment and production of iron-rich ULVZs in slab regions? It seems to me the latter can be true. This then is actually basically a novel idea, that water reaction with the core could produce iron-rich ULVZ material in cold subduction regions, perhaps even in the absence of partial melting. I believe such a discussion would improve the impact and significance of these findings quite strongly.

5. The authors cite the work of Otsuka and Karato (2012) (ref. 8). However, in discussion of the results (e.g., line 133) where the authors find penetration of molten iron into FeO-rich (Mg,Fe)O, the authors do not compare their results to the earlier publication. There should be some comment on whether the findings agree with the previous work, which also observed molten iron penetration into (Mg,Fe)O. In particular, is the length-scale of penetration similar or greater than in Otsuka and Karato? If it is higher here due to water, that point should be made. If it is similar, then the point should be clear that a mechanism for molten iron penetration has previously been demonstrated. The key difference here is that such iron enrichment can be promoted by the presence of water even in situations where (Mg,Fe)O grains are likely to represent only a small fraction of the bulk rock and thus may be unlikely to interconnect and promote iron penetration / enrichment at the mantle base without the presence of water.

6. Possible iron disproportionation and the role of nickel: The authors mention that the presence of iron metal in the silicate matrix for experiments where the iron was solid and not molten is evidence for disproportionation of iron in the silicate ($\text{Fe}^{2+} \rightarrow \text{Fe}^{3+} + \text{Fe}^0$). However, in Fig. 1, there is a faint signature of nickel in the silicate matrix in identical locations as where iron enrichment (perhaps as metal, though this is not obvious with oxygen and magnesium maps). The amount of nickel visually seems to be lower than in the case of molten iron, but it is not really clear given that the color scheme of the composition maps can be somewhat arbitrary. Can the authors comment on the presence of nickel in the silicate in Fig. 1e? And could Ni be implicated in the reaction? What about the production of NiO and NiH and possible solid solution of FeO with NiO? Have the authors looked into any possible literature on such an idea? This could be an interesting point to speculate on.

Response to review of the manuscript “Extensive iron–water exchange at Earth’s core–mantle boundary explains seismic anomalies”

The reviewer’s comments are shown in black, and our responses are in red.

Response to Referee #1

> Kawano et al. performed the high P-T experiments at 25-27 GPa and 1473-2473K under hydrous and anhydrous conditions to try explaining ULVZs at CMB. I like the idea, but their implication strongly relies on overmuch estimation, assumption and incorrect citation, e.g., extrapolating the experimental data at 25 GPa to CMB (~135 GPa), partitioning coefficient of Fe between post-perovskite and ferropericline, V_p and V_s of Brg (or PPv). I cannot recommend this manuscript for publication in Nature Communications. My comments/suggestions see the below:

Reply: We sincerely appreciate the valuable comments concerning this manuscript. The manuscript has been revised following the referee’s comments to convince all readers of our conclusions. We apologize for the confusion caused by the incorrect citation of Zhang et al., 2016 GRL, instead of the correct reference Zhang et al., 2016 EPSL. The correct reference, which we have now included in the reference list, uses data directly relevant to CMB conditions. Our original calculations were also based on accurate values for CMB pressures (~136 GPa), and we did not rely on extrapolation from 25 GPa. This issue arose simply due to a citation error in the reference list.

> One of main assumptions is that the reactions of $3\text{Fe} + \text{H}_2\text{O} = \text{FeO} + 2\text{FeH}$ and $\text{FeO} + \text{MgSiO}_3 = (\text{Mg,Fe})\text{O} + (\text{Mg,Fe})\text{SiO}_3$ must happen when water meets Earth’s core. But, the exsolved ferrous (Fe^{2+}) could undergo disproportionation ($\text{Fe}^{2+} \rightleftharpoons \text{Fe}^0 + \text{Fe}^{3+}$) at CMB conditions. If that, how does Fe^{3+} affect your assumed reaction? Can it still efficiently work on ‘Fe diffusion’ to Brg or Fp?

Reply: Disproportionation reactions occurred partially after the diffusion of FeO when MgSiO_3 reacted with the FeO component. Therefore, Fe^{3+} does not dominate the iron diffusion itself in the system. Moreover, as shown in additional supplementary Table 2, we observed (Mg,Fe)O in the recovered sample, which should contain ferrous iron even after diffusion. Although we do not deny the possibility of some influence on the kinetics, we consider it certain that partial disproportionation reactions do not strongly prevent Fe diffusion, as evidenced by the rapid iron movement (Fig. 1,2b).

> Nishi et al. (2017) experimentally showed pyrite-type FeOOH stabilizing at the CMB conditions, which is confusing to me why the authors claimed $\text{Fe} + \text{H}_2\text{O} \rightarrow \text{FeO} + \text{FeH}$, rather than FeO_2H_x in this study. Actually, the reaction of $\text{FeO} + \text{H}_2\text{O} \rightarrow \text{FeO}_2\text{H}_x + \text{FeH}_x$ should be the result, which has been studied and used to explain ULVZs (e.g., Liu et al., 2017 Nature). Therefore, ‘the idea’ is not new anymore. Again, the experimental conditions in this study (25-27 GPa) are far away from the stable field of FeO_2H_x ($> \sim 90$ GPa), where the authors tried to make the implication. How is possible to tell a story happening at CMB whereas using the data from > 2000 km shallower?

Reply: We carefully considered the reviewer’s comments and realized that we should have explained and discussed the problems of FeO_2H_x formation at ULVZs more thoroughly in the original manuscript. Many previous studies on this phase can confuse the actual chemical reactions at the CMB. To avoid confusion related to FeO_2H_x , we provided details of Nishi et al. (2020) in lines 71–76 of the revised manuscript.

First, in contrast to Liu et al. (2017), Nishi et al. (2017) did not show the stability of pyrite-type FeOOH (or FeO_2H_x) at the CMB due to the release of H_2O at high temperatures. Instead, they demonstrated the incorporation of hydrogen into the outer core via the reaction $\text{FeO} + \text{H}_2\text{O} \rightarrow \text{FeO} + \text{FeH}$, as stated in their Abstract (Nishi et al., 2017).

Additionally, experimental results at CMB pressure conditions using a diamond anvil cell (DAC) (Nishi et al., 2020, GRL) clearly indicated the formation of FeO instead of FeO_2H_x under realistic water concentrations, even within the stability field of FeO_2H_x above 70 GPa. As far as we know, FeO_2H_x can only appear under H_2O -saturated conditions and never coexist with metallic pure iron (not FeH_x) (see Fig. 3 in Nishi et al., 2020). Thus, we consider that the reaction $\text{FeO} + \text{H}_2\text{O} \rightarrow \text{FeO} + \text{FeH}$ should be applicable at the CMB, where an unlimited availability of iron and a limited amount of water are expected.

In the present study, we have uncovered a novel reaction mechanism that facilitates the activation of chemical interactions between minerals and iron under pressure, aided by the presence of minute quantities of water. As written in lines 157–163 and 133–139 of the revised manuscript, this mechanism includes not only FeO formation but also chemical partitioning, rapid diffusion paths of Fe, and liquid Fe penetration. This mechanism addresses the longstanding issue of the reaction kinetics within ULVZs.

> Their estimations of seismic velocities of Brg (or PPv) could be totally wrong in Fig. 4c,d and Table S3. Their velocity result fully depends on the extrapolation of sound velocities data of olivine at the extremely low P-T conditions compared to the CMB’s (Zhang et al., 2016). I cannot imagine how big error at their estimations could be because of the different mineral phases, different P-T conditions. It would be much more

convincing if the authors could provide the experimentally measured velocity data of Brg+Fp at their targeted P-T conditions.

Reply: We apologize for the misleading citation of Zhang (2016, GRL) instead of the correct reference Zhang et al. (2016, EPSL). In our manuscript, we calculated the sound velocities using data for bridgmanite and post-perovskite at CMB conditions. We did not use data from olivine at lower pressures.

> Lines 299-300: “D is the partition coefficient of FeO between post-perovskite and ferropericlase ($(\text{Fe}/\text{Mg})_{\text{ppv}} / (\text{Fe}/\text{Mg})_{\text{fp}}$), fixed at 0.4 in the calculation.” In general, D ranges at ~0.4-0.9 at lower mantle P-T conditions (Irifune et al., 2010; Prescher et al., 2014; Sinmyo and Hirose, 2013), but a mantle heavily enriched in Fe²⁺ would have a D of <0.1 (Nakajima et al., 2012). The implication of this ms actually fits better with the second case – D should be much lower than 0.4. What will the results be if the authors apply D of 0.1 or 0.9? Still matching the volume of ULVZs?

Reply: Based on the reviewer’s suggestion and further consideration, we recalculated our results using a D value of 0.2, which we now recognize as more appropriate for our study. Consequently, we have revised the manuscript (lines 279–282) and Figure 4 to reflect this change. In addition, we performed calculations using D values of 0.05 and 0.5 to explore the sensitivity of our results to variations in FeO partitioning. The results of these additional calculations have been included in Supplementary Fig. 5. As shown in these figures, variations in FeO partitioning within this range do not significantly impact our conclusions. This is because the FeO component reduces the seismic velocities of both postperovskite (or bridgmanite) and ferropericlase.

> Lines 109-111: What is the effect of pressure on the rim thickness of the FeO-rich layer? Will it become thinner or thicker at the CMB conditions than your estimation from the experimental conditions at the topmost lower mantle?

Reply: As the formation of FeO requires the reaction between Fe and H₂O in our model, the thickness of the FeO-rich layer in our experiments is controlled by the amount of water supplied at the interface rather than by the pressure, as shown in lines 118–126 and in Fig. 2a of the revised manuscript. Since the volume of ULVZ can be roughly determined from seismological observations (lines 277–278), we can estimate the water content assuming a fixed volume of ULVZs (Fig. 4 and supplementary Figs. 5 and 6). Through this estimation, the V_p , V_s , and ρ of minerals at the CMB condition were used. The extremely active reaction mechanisms (spanning over a few kilometers as estimated

from the diffusion coefficient at the CMB in the previous studies) found in this study make the above estimation possible (see discussion in lines 127–137). This mechanism represents a significant advancement from previous studies, which only showed the phase relation (e.g., Nishi et al., 2022 GRL; Liu et al., 2017 Nature; Nishi et al., 2017 Nature).

Lines 112-114: If I understand Table S1 correctly, T does significantly affect the thickness of the FeO-rich layer, which is different with the description from the authors.

Reply: Thank you for pointing this out. We added the sentence “when iron is solid” in line 123 in the revised manuscript. The description was applicable at temperatures where iron is solid. The discussion is not reasonable at higher temperatures where iron is molten because iron penetration adittionary occurs.

Response to referee#2

>Kawano and co-authors present a series of experiments on bridgmanites interacting with iron(-nickel) metal. By using identical experiments that directly compare the effect of water in bridgmanite, as well as the effect of solid or liquid iron metal, the authors are able to make inferences on how water may affect reactions at Earth's core-mantle boundary. In particular, they show evidence that solid anhydrous bridgmanite will not react with iron, while the presence of water seems to induce the formation of FeO and FeH through reaction of water with iron. The production of FeO in the form of iron-rich (Mg,Fe)O seems to further promote diffusion of iron metal that otherwise would not occur. By varying the amount of water (or rather, volume of the hydrous bridgmanite sample), the authors demonstrate that the extent of FeO formation and iron penetration into the silicate is controlled by the amount of water available for reaction. Thus, the presence of water may promote iron enrichment of the mantle base, providing a plausible origin scenario for observed seismic structures like ultralow velocity zones that could be explained by iron-rich rock.

>I believe this article would be well suited for publication in Nature Communications after several aspects are addressed and revised. Previous experiments have considered some similar reactions, especially looking at anhydrous silicates, anhydrous oxides, and less realistic water-saturated scenarios. This work is novel in its direct comparison of hydrous with anhydrous silicate material, which demonstrates a clear difference in rock-metal interaction and does so for a relatively realistic water concentration. Repetition of experiments and reproducibility of the findings is assuring to see. The scientific topic is an active hot topic in Earth science, with many open questions and ongoing work on the nature of the mantle-core interface and the role of water, especially in relation to heterogeneous mantle base structures that are actively being discovered by seismic studies. This work is a timely and valuable contribution to these discussions that I expect will motivate further experiments, especially at higher pressures. I identify a few key areas that need revision, particularly on showing more data, including some missing literature references, and reworking the interpretation for sound velocities, as well as some questions that need to be clarified by the authors before the work can be acceptable for publication.

We thank the referee for highlighting the important points of our paper. Moreover, we thank you for the valuable comments and careful check of this manuscript. We found the

comments below very helpful and have revised the manuscript accordingly.

1. Chemical analysis

(a) I am very interested to see more elemental maps reported. I do not see any maps for Mg or O. This makes it difficult to assess the claim that one single phase of iron-rich (Mg,Fe)O is being produced along with Fe metal. This is very important because this claim is the central point of the paper. How can the authors be sure that the bright grains in Fig. 1 are iron metal alloy and not some other iron-rich phase? Can the authors provide the full set of elemental maps that were collected for the heating runs that are shown in Fig. 1? What about for other experiments, such as the one shown in Table S2? It would be helpful to see all the elemental maps for that experiment (2829). See more on this below (comment 1d).

Reply: We agree with the referee's suggestion that EDS maps for Mg and O are useful for identifying the phases for readers. We have revised Fig. 1 to increase the resolution and added maps for Mg and O in the revised manuscript. Additionally, we have included the maps in the supplementary Figs. 3 (OT2842) and 4 (OT2829). From supplementary Fig. 4, the presence of Fe metal, (Mg,Fe)SiO₃, and (Mg,Fe)O phases can be easily recognized. Through this revision, we have replaced the EPMA maps with the EDS maps obtained using FE-SEM (as written in lines 253–256 of the revised manuscript), as the latter had provided better data.

(b) In addition, I am curious to hear discussion on hydrous Fe-O-H phases that have been previously reported (e.g., ref 14 – Liu et al. 2017 Nature) and to have the authors comment on whether they think the production of such phases is possible in these samples. In other words, without the use of XRD, how can the authors be sure that the new phases formed are iron metal alloy and iron-rich (Mg,Fe)O and not some other phases? How can the authors be sure that phases like exotic Fe-O-H phases and FeSi (e.g., Knittle and Jeanloz 1991 Science, Fu et al. 2023 Nature) are not being formed?

Reply: We consider that the Fe-O-H phase, reported in the previous studies, is unrealistic to appear at CMB because this phase is easily reduced by metallic Fe. Experimental results at CMB pressure condition using diamond anvil cell (DAC) (Nishi et al., 2020 GRL) clearly indicated the formation of FeO instead of FeOOH (or FeO₂H_x) under realistic water concentration even in the stability field of pyrite-type FeOOH above 70 GPa. At lower pressures where ϵ -FeOOH (a low-pressure polymorph of pyrite-type FeOOH) is stable, FeOOH is reduced by Fe to form FeO (Figure S4 in Nish et al., 2020 GRL). In our samples, the elemental maps are useful to detect FeOOH because this phase does

not contain Mg. As shown in Supplementary Figure 4 and Supplementary Table 2 (the most likely condition to appear FeOOH because of the low temperature), the FeO-rich phase contains Mg (this phase should be ferropericlase). Additionally, all elemental maps (Figs. 1e, 1d, supplementary figs. 3, 4) provided in the revised manuscript suggest that there is no FeSi phase in the FeO-rich layer.

(c) My understanding is that EPMA should have some finite activation volume in the sample usually at least ~1 μm ? This would mean that the probe is sensitive to a volume of sample that likely includes grains deeper inside the sample below the surface. Are the uncertainties on compositions reported in Table S2 realistic?

Reply: We have reanalyzed and added the compositions of bridgmanite and ferropericlase from runs OT 2829, 2933b, and OT 2834b in Supplementary Table 2. For compositional analysis, we used EDS by FE-SEM instead of WDS by EPMA to reduce the overlap in compositional analysis. The uncertainties in the composition of bridgmanite are realistic because its grain size ($>3 \mu\text{m}$) is large enough to analyze quantitatively. However, despite our efforts, a small overlap with other grains was observed in ferropericlase. We added the sentence “Note that the minute amount of SiO_2 and Al_2O_3 components appearing in ferropericlase are likely due to overlap with bridgmanite” in the caption of Supplementary Table 2.

(d) Is there a reason that compositional data is provided for only one sample? It would be very helpful to see compositions for multiple samples, especially those shown in Fig. 1. This would clarify how consistent the resulting compositions of (Mg,Fe)O and bridgmanite are. In particular, does the iron content of the (Mg,Fe)O phase that is produced depend on the extent of the reaction / amount of water? Further, can the authors report an Fe partitioning coefficient between bridgmanite and magnesiowüstite across the range of samples for which compositions are measured? How do the calculated coefficients compare to previous work by Dorfman et al. (2021) Minerals?

Reply: We have reanalyzed and added the compositions of bridgmanite and ferropericlase from runs OT 2829, 2933b, and OT 2834b in Supplementary Table 2. Although we made an effort to conduct further chemical analysis, it was difficult to obtain meaningful data from the other runs because of the small grain size of ferropericlase. The obtained partition coefficients ($D = 0.05\text{--}0.22$, supplementary Table 2) are roughly consistent with those previously reported by Dorfman et al. (2021). We recalculated our results using a D value of 0.2 (we used 0.4 previously), which we now recognize as more appropriate for our study. Also, we performed additional calculations using D values of

0.05 and 0.5 to explore the sensitivity of our results to variations in FeO partitioning (Supplementary Fig. 5 and lines 280–282 of the revised manuscript). These different D values do not impact our conclusions, and our conclusion remains the same: extensive iron-water interaction can account for the formation of ULVZs.

2. Missing references

(a) The authors motivate the paper by stating that iron enrichment of mantle minerals is a strong candidate for ULVZs (lines 44-47). However, there are no references included for this in the introduction or anywhere else. There is a whole body of recent work about iron enrichment in ULVZs that is completely absent from the manuscript. This should be corrected in the introduction, which I will comment on here, and then plays into the analysis of sound velocities at the end of the paper, which I will comment on below.

Papers that should be included at minimum: Wicks et al (2010) GRL; Wicks et al. (2017) GRL; Lai et al (2022) G3. These reported low sound velocities for iron-rich (Mg,Fe)O (Wicks) and demonstrated that such FeO-rich solid rocks can explain ULVZ velocities (Lai). Other papers to be considered are Dobrosavljevic et al. (2019) Minerals (effect of iron content and methods of sound velocity modeling) and Jackson and Thomas (2021) AGU Monograph (broad range of ULVZ locations).

Reply: We agree with the referees' comments regarding the missing references on iron enrichment in mantle minerals as a strong candidate for ULVZs. These papers clearly support our research motivation, demonstrating that iron enrichment in mantle minerals is a strong candidate for ULVZs. We have included key references (Wicks et al., 2010 GRL; Wicks et al., 2017 GRL; Lai et al., 2022 G3; Dobrosavljevic et al., 2019) in lines 45–48 in the revised introduction as follows:

Many studies have shown that iron enrichment in mantle minerals is a significant factor contributing to the properties of ULVZs. For instance, it has been demonstrated that iron-rich (Mg,Fe)O exhibits low sound velocities⁴⁻⁷. The diminished seismic wave velocities observed in FeO-rich mantle minerals quantitatively align with those of ULVZs^{8,9}.

(b) Other mechanisms have been proposed for production of iron enrichment at the mantle base. These include in particular a basal magma ocean (Labrosse et al. 2007 Nature) and subduction of banded iron formations (Dobson and Brodholt 2005 Nature). These should at least be mentioned in the introduction as another possible way to further enrich the mantle base in iron, beyond the mechanism proposed in this study. Authors should also include Knittle and Jeanloz (1991) Science as an earliest experiment proposing iron enrichment at the CMB due to reaction of silicates with the core.

Reply: As per your suggestion, we have included references to the basal magma ocean model (Labrosse et al., 2007) and the subduction of banded iron formations (Dobson and Brodholt, 2005) in lines 49–51 of the revised manuscript as follows:

Several hypotheses have been proposed to explain the origin of iron-rich ULVZs, including a basal magma ocean¹⁰ and the subduction of banded iron formations¹¹, both of which involve the increase in iron content.

We have also incorporated the reference to Knittle and Jeanloz (1991), who conducted pioneering experiments proposing iron enrichment at the CMB due to reactions between silicates and the core (line 55).

3. Sound velocity calculations

(a) The choices made for sound velocity calculations are not very clear to me. I understand the point about mass balance due to differing amounts of water. However, the elastic properties used for (Mg,Fe)O are listed only as those calculated computationally for pure MgO and for (Mg_{0.75}Fe_{0.25})O. This is a much lower iron content than what the authors themselves found experimentally, where the experimental reaction produced an (Mg,Fe)O composition with 65% Fe. This experimental value is much closer to the iron-rich (Mg,Fe)O compositions proposed previously to explain ULVZ signatures (see reference list in my comment 2), such as iron levels of 75%-90%. In addition, those compositions have experimentally determined sound velocities and densities. So it is not clear to me why those values are not being used for example calculations. It's further not clear what the compositions of (Mg,Fe)O are in the calculations for high amounts of water content. Do these exceed the Fe=25% level? It seems they must given how low the calculated bulk sound velocities are. In that case, it is likely that a linear extrapolation from MgO and (Mg_{0.75}Fe_{0.25})O is no longer valid at higher Fe contents. Our community does not have a good understanding, if any, of iron content effect on sound velocities at intermediate compositions between Fe~25% and Fe~75%. If the authors want to model iron-rich (Mg,Fe)O sound velocities above the Fe=25% compositions, then I believe they should in some way use the experimentally reported sound velocities for very iron-rich (Mg,Fe)O as in the references above. See a suggestion below in comment 3c.

Reply:

We have revised the manuscript to be clearer regarding the elastic wave velocity calculation in Fig. 4. In our calculation, the composition of (Mg,Fe)O and (Mg,Fe)SiO₃ change as a function of input water, as shown in the original supplementary Fig. 3. Based on the reviewer's comments, we recognized that this figure is very important for clarifying our calculation model, and we have incorporated it into Fig. 4b, in the revised

manuscript.

We totally agree with the reviewer's consideration. However, as shown in Fig. 4b (originally Supplementary Fig. 3), the required compositions of (Mg,Fe)O are continuous, with Fe/(Mg+Fe) ratios ranging from 0.2 to 0.6. Although it is known that the FeO component reduces the V_p/V_s value for ferropericlasite, experimental data are missing for such specific values, as the reviewer pointed out.

Since previous experimental data do not cover the required compositions, we have used the reference values taken from the theoretical calculations in the previous studies, and a linear extrapolation for different Fe components was performed (in lines 293-298 of the revised manuscript and revised supplementary Table 3). Fortunately, this estimate roughly reproduces the V_p and V_s obtained from experiments with a composition of $\text{Fe}_{0.78}\text{Mg}_{0.22}\text{O}$, which is the closest available experimental data to the required composition range (see the figure below and supplementary Fig. 6). Thus, we believe this approach is the most appropriate given the current limitations in experimental data, and it ensures the reliability of our sound velocity estimations.

Even if we used largely different values of sound velocities, our model to explain ULVZs remains valid because the effect of iron lowering sound velocity is consistent in all experimental and theoretical studies. Nevertheless, as written in lines 190–192 of the revised manuscript, further experimental studies on sound velocity measurements of ferropericlasite with different Fe components, combined with seismological studies related to ULVZ volume fractions, will contribute to better water quantification.

(b) Further, it is not clear from the methods how exactly the sound velocities are calculated. The authors state in line 308 that “Linear compositional dependence on the elastic parameters of given phases was assumed.” There is not enough detail provided. Is this being done for elastic moduli? What assumptions are being made for stress and strain partitioning? Calculating velocities of bulk rock requires decisions on how to average

elastic moduli, as in the common Voigt and Reuss bounds or the Voigt-Reuss-Hill average. Authors may refer to calculations detailed in e.g., Dobrosavljevic et al., (2019) Minerals and Lai et al. (2022) G3. This point is especially important for rocks comprised of minerals with very different elastic moduli, as in bridgmanite and iron-rich (Mg,Fe)O. In such a case, the difference between Voigt and Reuss bounds (or uncertainties on V-R-H averages) can be incredibly large (again, refer to e.g., Lai et al. (2022) G3). These details need to be explicitly stated at least in the supplementary material if not in the Methods.

Reply: We acknowledge that the original manuscript did not provide sufficient detail on how the sound velocities were calculated. To address this, we have expanded the Methods section (lines 265–267, 278–282, 297–299) and the supplementary material to include a more detailed explanation. Velocities of bulk rock are made by Voigt-Reuss-Hill average as described in line 299 of the revised manuscript. (We used Voigt bounds in the original calculations.). As written in the previous reply, the required compositions of (Mg,Fe)O are continuous, with Fe/(Mg+Fe) ratios ranging from 0.2 to 0.6. We have used the reference values taken from the theoretical calculations in the previous study (Table S3) and a linear extrapolation for different Fe components was performed. While we did not use directly the experimental values suggested in the referee's comments for our calculations, we have cited these papers in lines 46–48 and 297–299 to provide context and support for our approach.

(c) It seems to me that the goal of the authors is to show that reasonable amounts of subducted water can produce a high level of iron enrichment through reaction with the core and explain ULVZ properties. In light of my comments above, I think a stronger approach (or at least a strong complement to the current approach) would be to graphically show how much iron enrichment in (Mg,Fe)O is produced due to such a reaction. If the authors can plot (Mg,Fe)O composition and concentration of (Mg,Fe)O in the bulk rock as a function of reacted water, then the authors could simply refer to existing literature on FeO-rich rocks as an explanation for ULVZs (see reference list in my comment 2 above). This would be a much cleaner message in the discussion and in Figure 4 without needing to go into the many details of sound velocity modeling done in previous work. Authors can also consider plotting the x-axis as units of fraction of ocean mass instead of in units of kg (as mentioned in line 168), which are hard to interpret.

Reply: We appreciate the careful consideration of our study and the constructive comments provided. The figure illustrating the composition as a function of reacted water was already included in supplementary Fig. 3 of the original manuscript. Recognizing the

importance of this figure, we have now moved it to Figure 4b in the revised manuscript. This adjustment aims to enhance the clarity of our discussion. As per your suggestion, we have modified the units on the X-axis from kilograms (kg) to the fraction of ocean mass. We also added the sentence “The mass of seawater corresponds to 1.38×10^{21} kg.” in the caption of Figure 4.

4. Discussion around CMB temperature and melting in the context of ULVZs: recent seismic studies are increasingly finding evidence for ULVZ existence in areas of active subduction where temperatures are presumed to be relatively low (e.g., Su et al. 2024 Nature Geoscience, Wolf et al. 2024 Nature Geoscience). I think these studies actually strengthen the findings of this paper. Instead of focusing the discussion so much on high temperature and partial melting, the authors could consider discussing how their findings actually provide a compelling mechanism to create iron enrichment in regions of subducted slab debris by reaction of hydrous silicate with core fluid. My question to the authors here: do the experimental findings actually require higher temperatures to promote enough reaction for iron enrichment? Or can the authors apply the results even to colder regions similarly well to give a viable mechanism for CMB iron enrichment and production of iron-rich ULVZs in slab regions? It seems to me the latter can be true. This then is actually basically a novel idea, that water reaction with the core could produce iron-rich ULVZ material in cold subduction regions, perhaps even in the absence of partial melting. I believe such a discussion would improve the impact and significance of these findings quite strongly.

Reply: We thank the reviewer for insightful comments and suggestions and for referencing the recent seismic studies by Su et al. (2024) and Wolf et al. (2024), which indeed provide compelling evidence for the existence of ULVZs in regions of hydrous plate subduction. As the reviewer pointed out, our model provides a reasonable explanation for the formation of iron-rich layers through the simple mechanism of water transport by subducted slabs and their reaction with the outer core. We have included these papers and perspectives in lines 169–172 of the revised manuscript, with the statement, "Seismological observations imply that some ULVZs are related to plates subducted to the deep lower mantle (Su et al., 2024; Wolf et al., 2024). This supports our hypothesis that iron-rich layers can form through the transport of water by subducted slabs and their reaction with the outer core."

However, we also recognize that incorporating partial melting is advantageous when considering the quantitative aspects of seismic wave velocities and reaction kinetics.

Therefore, we have retained the discussions on partial melting at high temperatures in the revised manuscript. By addressing both the simpler mechanism and the more quantitative approach involving partial melting, we believe our revised manuscript provides a comprehensive and balanced explanation for the formation of iron-rich ULVZs in different thermal regimes.

In response to the reviewer's comments, we have removed the sentence "ULVZs are primarily observed in relatively hot regions with large temperature gradients, such as near large low-velocity provinces, rather than in the coldest regions directly beneath subduction zones" from the revised manuscript, as this part is difficult to assert definitively.

5. The authors cite the work of Otsuka and Karato (2012) (ref. 8). However, in discussion of the results (e.g., line 133) where the authors find penetration of molten iron into FeO-rich (Mg,Fe)O, the authors do not compare their results to the earlier publication. There should be some comment on whether the findings agree with the previous work, which also observed molten iron penetration into (Mg,Fe)O. In particular, is the length-scale of penetration similar or greater than in Otsuka and Karato? If it is higher here due to water, that point should be made. If it is similar, then the point should be clear that a mechanism for molten iron penetration has previously been demonstrated. The key difference here is that such iron enrichment can be promoted by the presence of water even in situations where (Mg,Fe)O grains are likely to represent only a small fraction of the bulk rock and thus may be unlikely to interconnect and promote iron penetration / enrichment at the mantle base without the presence of water.

Reply: Although direct comparison is difficult due to the different experimental conditions and limited ferropericlasite fraction, we observed the iron enrichment with a thickness of $\sim 100 \mu\text{m}$, which is comparable to the previous study as written in lines 149–152 of the revised manuscript. Although we consider that the length should be limited by water mass, further experimental studies are required to conclude the reaction mechanism and kinetics in such a complex system.

6. Possible iron disproportionation and the role of nickel: The authors mention that the presence of iron metal in the silicate matrix for experiments where the iron was solid and not molten is evidence for disproportionation of iron in the silicate ($\text{Fe}^{2+} \rightarrow \text{Fe}^{3+} + \text{Fe}^0$). However, in Fig. 1, there is a faint signature of nickel in the silicate matrix in identical locations as where iron enrichment (perhaps as metal, though this is not obvious with oxygen and magnesium maps). The amount of nickel visually seems to be lower than in

the case of molten iron, but it is not really clear given that the color scheme of the composition maps can be somewhat arbitrary. Can the authors comment on the presence of nickel in the silicate in Fig. 1e? And could Ni be implicated in the reaction? What about the production of NiO and NiH and possible solid solution of FeO with NiO? Have the authors looked into any possible literature on such an idea? This could be an interesting point to speculate on.

Reply: We have added maps for Mg and O as well as Ni in Fig. 1e and 1f of the revised manuscript (In the revised manuscript, we have swapped Fig. 1e and Fig. 1f.). The maps show the presence of Ni in the metal. Additionally, compositional analysis (revised supplementary Table 2) has revealed that a small amount of Ni is present in ferropericlase. The Fe/Ni ratio in the ferropericlase is much lower than that in the starting metal material. We consider that when the Fe-Ni alloy is solid (Fig. 1e), the flux of Ni from the metal to the FeO-rich layer via the diffusion was limited due to the higher siderophilicity of Ni relative to Fe (e.g., Campbell et al., 2009 EPSL) as described in lines 146–147 of the revised version of the manuscript. On the contrary, when we utilised the Fe–Ni alloy as the starting material and melted it, its particles migrated into the FeO-rich layer while retaining their original composition. Based on our data and the absence of relevant literature, further discussion regarding Ni, such as the formation of Ni-H, is difficult.

REVIEWERS' COMMENTS

Reviewer #1 (Remarks to the Author):

Review of the revised manuscript by Kawano et al.

In the revised version, Kawano et al. carefully corrected the wrong citation in the original ms. for the calculation of sound velocities, which is a root for the implication of their study, and made a clear explanation as well in the reply letter. They also corrected the partition coefficient of FeO between post-perovskite and ferropericlasite ($(\text{Fe}/\text{Mg})_{\text{ppv}} / (\text{Fe}/\text{Mg})_{\text{fp}}$) from 0.4 to 0.2, which is reasonable to their experimental conditions. I would like to recommend this manuscript for publication in Nature Communications.

Reviewer #2 (Remarks to the Author):

I would like to thank Kawano and co-authors for their careful work revising the manuscript and addressing comments from myself and the other reviewer. I believe that all reviewer comments have been convincingly addressed and that this manuscript is well-suited for publication in Nature Communications. I include just a few small points and questions for the authors to consider and address before publication.

1. The topic of FeO_2Hx is something brought up by both reviewers, and the new discussion does a nice job addressing these details in a compelling way. In the rebuttal, the authors mention that neither epsilon FeOOH nor FeSi are observed in the reaction products, tested using Mg and Si maps. This is an interesting finding that I think is worth mentioning explicitly in the main text, maybe at the end of the second section (~line 115).

2. The addition of more EDS maps is very helpful and now clearly demonstrates the presence of both Fe metal and iron-rich $(\text{Mg},\text{Fe})\text{O}$ in the bridgmanite matrix. One suggestion and one comment here:

a. The authors could further consider combining Fig. S4 into Fig. 1 to make a large 2-column figure. Showing 1c, 1d, and Fig. S4 together in the main text would be even more compelling for readers to clearly see the different phases produced. Alternately, authors could even consider making both of these two separate main figures. There should be plenty of room for an additional figure in Nat. Comm.

b. I note in Fig. 1e the presence of regions in the quenched Fe-Ni liquid that appear to be something like FeO ? These are regions with no Ni and no Mg. Can the authors comment on this?

3. The explanation of sound velocity calculations and revisions made to the manuscript on this topic are nicely done. The authors included a figure in the rebuttal showing that modeled sound velocities for

iron-rich (Mg,Fe)O agree well with values extrapolated from experimental work, as reported eg in Lai et al. (2022). I would suggest the authors consider including this figure in the supplement. They already mention this agreement in the Methods text, and having this figure could be a nice additional illustration that supports the credibility of their approach for modeling sound velocities.

4. How is total ULVZ volume determined? I see in the Methods that an assumption is made of a 20 km ULVZ covering 12% of the CMB. This seems somewhat reasonable, but it would be nice to have a reference mentioned for this (maybe looking at Yu and Garnero 2018 G3?). It is likely that the true value could be higher due to limited seismic ray coverage of the CMB in existing observations (and active new discoveries of ULVZs at new locations). This would mean that the total water mass involved in the proposed reaction could be higher. Authors could consider mentioning this in the main text (maybe ~line 189).

Response to review of the manuscript “Extensive iron–water exchange at Earth’s core–mantle boundary can explain seismic anomalies”

The reviewer’s comments are shown in black, and our responses are in red.

Response to Referee #2

> I would like to thank Kawano and co-authors for their careful work revising the manuscript and addressing comments from myself and the other reviewer. I believe that all reviewer comments have been convincingly addressed and that this manuscript is well-suited for publication in Nature Communications. I include just a few small points and questions for the authors to consider and address before publication.

Reply: We are pleased to hear that the reviewer comments have been convincingly addressed and that the manuscript is well-suited for publication in Nature Communications. We appreciated your additional points and questions.

> 1. The topic of FeO₂Hx is something brought up by both reviewers, and the new discussion does a nice job addressing these details in a compelling way. In the rebuttal, the authors mention that neither epsilon FeOOH nor FeSi are observed in the reaction products, tested using Mg and Si maps. This is an interesting finding that I think is worth mentioning explicitly in the main text, maybe at the end of the second section (~line 115).

Reply: We have added this information at the end of the second section (~line 115). " As confirmed through elemental maps (Fig. 1), neither FeSi phase and hydrous FeOOH phases were observed."

> 2. The addition of more EDS maps is very helpful and now clearly demonstrates the presence of both Fe metal and iron-rich (Mg,Fe)O in the bridgmanite matrix. One suggestion and one comment here:

a. The authors could further consider combining Fig. S4 into Fig. 1 to make a large 2-column figure. Showing 1c, 1d, and Fig. S4 together in the main text would be even more compelling for readers to clearly see the different phases produced. Alternately, authors could even consider making both of these two separate main figures. There should be plenty of room for an additional figure in Nat. Comm.

Reply: We have combined Fig. S4 into Fig. 1 to create a comprehensive figure. This new figure arrangement now includes 1c, 1d, and new 1e, allowing readers to clearly see the

different phases produced.

> b. I note in Fig. 1e the presence of regions in the quenched Fe-Ni liquid that appear to be something like FeO? These are regions with no Ni and no Mg. Can the authors comment on this?

Reply: We have added the following sentence to the manuscript to clarify this point in the caption of Fig.1 : "The quenched Fe-Ni liquid forms metal dendrites due to the separation of FeO." This indicates that the FeO regions observed are a result of the separation process during quenching.

> 3. The explanation of sound velocity calculations and revisions made to the manuscript on this topic are nicely done. The authors included a figure in the rebuttal showing that modeled sound velocities for iron-rich (Mg,Fe)O agree well with values extrapolated from experimental work, as reported eg in Lai et al. (2022). I would suggest the authors consider including this figure in the supplement. They already mention this agreement in the Methods text, and having this figure could be a nice additional illustration that supports the credibility of their approach for modeling sound velocities.

Reply: We agreed with the reviewer's suggestion and have added the Supplementary Fig.6 illustrating the agreement between the modeled sound velocities for iron-rich (Mg,Fe)O and the experimental values from Lai et al. (2022).

> 4. How is total ULVZ volume determined? I see in the Methods that an assumption is made of a 20 km ULVZ covering 12% of the CMB. This seems somewhat reasonable, but it would be nice to have a reference mentioned for this (maybe looking at Yu and Garnero 2018 G3?). It is likely that the true value could be higher due to limited seismic ray coverage of the CMB in existing observations (and active new discoveries of ULVZs at new locations). This would mean that the total water mass involved in the proposed reaction could be higher. Authors could consider mentioning this in the main text (maybe ~line 189).

Reply: We have included a reference to Yu and Garnero (2018 G3) and Dobson and Brodholt (2005) in the Methods section to provide a basis for the assumption. Additionally, we have mentioned the quantification of water contents, noting that seismological studies related to ULVZ volume fractions including the discovery of new ULVZs will contribute to better water quantification, in lines 191–193 of the revised manuscript.